# Deep Integration of Fiber-Optic Communication and Sensing Systems Using Forward-Transmission Distributed Vibration Sensing and on–off Keying

**DOI:** 10.3390/s24175758

**Published:** 2024-09-04

**Authors:** Runlong Zhu, Xing Rao, Shangwei Dai, Ming Chen, Guoqiang Liu, Hanjie Liu, Rendong Xu, Shuqing Chen, George Y. Chen, Yiping Wang

**Affiliations:** 1Shenzhen Key Laboratory of Ultrafast Laser Micro/Nano Manufacturing, Guangdong and Hong Kong Joint Research Centre for Optical Fibre Sensors, Shenzhen University, Shenzhen 518060, China; 2300453015@email.szu.edu.cn (R.Z.); 2250453012@email.szu.edu.cn (X.R.); 2350453003@email.szu.edu.cn (S.D.); 2210452010@email.szu.edu.cn (M.C.); 2020274017@email.szu.edu.cn (G.L.); liuxiaoniu@whut.edu.cn (H.L.); ypwang@szu.edu.cn (Y.W.); 2Key Laboratory of Optoelectronic Devices and Systems of Ministry of Education/Guangdong Province, State Key Laboratory of Radio Frequency Heterogeneous Integration, College of Physics and Optoelectronic Engineering, Shenzhen University, Shenzhen 518060, China; 3Ocean College, Zhejiang University, Hangzhou 316000, China; xurd@zju.edu.cn; 4Jiangsu Ocean Technology and Equipment Innovation Center, Suzhou 215000, China; 5International Collaborative Laboratory of 2D Materials for Optoelectronics Science and Technology of Ministry of Education, Institute of Microscale Optoelectronics, Shenzhen University, Shenzhen 518060, China; shuqingchen@szu.edu.cn; 6Guangdong Laboratory of Artificial Intelligence and Digital Economy (SZ), Shenzhen 518107, China

**Keywords:** communication–sensing integration, fiber-optic, distributed sensor, forward transmission, long distance, OOK, on–off keying

## Abstract

The deep integration of communication and sensing technology in fiber-optic systems has been highly sought after in recent years, with the aim of rapid and cost-effective large-scale upgrading of existing communication cables in order to monitor ocean activities. As a proof-of-concept demonstration, a high-degree of compatibility was shown between forward-transmission distributed fiber-optic vibration sensing and an on–off keying (OOK)-based communication system. This type of deep integration allows distributed sensing to utilize the optical fiber communication cable, wavelength channel, optical signal and demodulation receiver. The addition of distributed sensing functionality does not have an impact on the communication performance, as sensing involves no hardware changes and does not occupy any bandwidth; instead, it non-intrusively analyzes inherent vibration-induced noise in the data transmitted. Likewise, the transmission of communication data does not affect the sensing performance. For data transmission, 150 Mb/s was demonstrated with a BER of 2.8 × 10^−7^ and a *Q_dB_* of 14.1. For vibration sensing, the forward-transmission method offers distance, time, frequency, intensity and phase-resolved monitoring. The limit of detection (LoD) is 8.3 pε/Hz^1/2^ at 1 kHz. The single-span sensing distance is 101.3 km (no optical amplification), with a spatial resolution of 0.08 m, and positioning accuracy can be as low as 10.1 m. No data averaging was performed during signal processing. The vibration frequency range tested is 10–1000 Hz.

## 1. Introduction

The development of distributed fiber-optic sensors has made great strides over the past few decades, allowing for real-time monitoring of cities, agriculture, actional infrastructure and remote environments [1]. The information can be collected and reported in real time to provide critical services such as early warnings of earthquakes [2,3,4]. However, the high cost of deploying new optical fiber in remote places, especially across vast oceans, restricts its vast potential for contributing to the Internet of Things (IoT). Fortunately, fiber-optic communication is already well established, with billions of kilometers of fiber installed globally and more than 1.4 million km installed at the bottom of the oceans for transoceanic internet networking. Fiber-optic sensors already borrow heavily from fiber-optic communication in terms of optical fiber, components, lasers, detectors and even demodulation schemes. In the pursuit of a cost-effective solution to the need for local/global information network, deep integration between distributed sensing and communication is highly desirable, and in a way inevitable.

Many attempts have been made in recent years to incorporate distributed sensing functionality into existing fiber-optic communication systems, notably the addition of phase-sensitive optical time-domain reflectometry (or distributed acoustic sensing (DAS)) [5,6,7]. Although this approach has realized the integration to some extent [8], the different characteristics of forward-transmitted light (communication) and Rayleigh backward-scattered light (DAS) mean that the processing requirements are very different. For example, the inclusion of DAS requires additional hardware and new optical fiber, and the single-span sensing distance of backscattered light is shorter than that of forward-transmitted light (typically ~50 km without inline amplification), meaning that the optical amplifiers (repeaters) used by communication networks, separated by distances of up to 90 km, are incompatible with DAS.

Forward-transmission-based distributed fiber-optic sensors [9] are an emerging technology that uses forward-propagating light instead of backscattered light, which leads to a much higher signal-to-noise ratio and a longer single-span sensing distance (~200 km [10]). The downside is the difficulty of resolving the fiber length under perturbation, as well as distinguishing multiple vibration points of the same frequency. Nonetheless, such sensing schemes are compatible with deep integration into fiber-optic communication, which means sharing the same substrate (e.g., optical fiber), the same channel (e.g., wavelength), the same signal (e.g., portion of light) and the same demodulator (e.g., coherent receiver), with absolutely no hardware changes (which is thus very cost-effective). Furthermore, this type of distributed sensing has no impact on communication and does not occupy bandwidth. Different types of demodulation schemes for sensing have been reported thus far, including intensity [11], phase [12,13] and polarization [14]. Some research groups have reported using fiber-optic communication hardware for distributed sensing and have demonstrated ultra-long multi-span sensing distances, up to thousands of kilometers long, using existing submarine communication cables [15,16,17]. However, the sensing aspect lacks vibration positioning functionality and there are no experiment results on nor analysis of the communication aspect for studying the impact of vibration on data transmission, and vice versa. In 2022, Ezra Ip et al. [18] published promising research results aiming towards such deep integration (53.9 km single span, 5 km positioning accuracy). However, the sensing part lacked true distributed sensing functionality, as it can only accurately measure one vibration point at a time. In addition, their work mainly focused on large-capacity communication, and lacked detailed analysis of the sensing performance, as well as missing key metrics such as sensitivity, limit of detection, etc.

In this work, we present a form of deep integration between forward-transmission-based fiber-optic distributed vibration sensing and fiber-optic communication. To the best of our knowledge, this is the first reported study of forward-transmission distributed fiber-optic sensing within the infrastructure of fiber-optic communication, which analyzes their respective performances and mutual impact on each other. In addition, we report on the longest single-span sensing distance (~100 km) for fiber-optic communication–sensing integration, a distance that encompasses most communication repeater distances (usually up to 90 km).

As a proof of demonstration, the simplest form of amplitude shift keying (ASK) is used, namely on–off keying (OOK). This type of deep integration does not need hardware changes, nor does it take up bandwidth for distributed sensing, as the nature of light propagation and optical amplification is identical to those of optical communication. The distributed sensing aspect, based on intensity demodulation, fully utilizes the optical communication path, pulse code modulation (PCM) and hardware, and demonstrates multi-point vibration positioning. The impact of the two functionalities on each other when both are operating is studied in terms of bit error rate (BER), limit of detection (LoD) and sensing distance. The results provide the groundwork for the deep integration of fiber-optic communication–sensing systems, which will pave the way for the deployment of a large-scale oceanic information network that can monitor earthquakes, tsunamis and ship traffic, and can gain new insights into our ever-changing natural environment.

## 2. Integrated System Setup

In a deeply integrated system between communication and sensing, the communication infrastructure is used as the carrier for sensing, while the sensing add-on is minimalistic and non-intrusive. For example, when resolving vibrations along the communication fiber, although the vibrations naturally impact the quality of data transmission, the sensing aspect only analyzes the digital data received and thus does not affect data transmission in terms of hardware changes and bandwidth. Vibration-induced noise has always been a part of data transmission in communication, and clearly any problems associated with such noise have already been solved.

The operational concept of communication–sensing integration is illustrated in Figure 1. Note that even for intensity-based communication, coherent receivers have been used [19,20], due to their versatility in supporting several demultiplexing schemes, such as intensity and phase, which can increase the overall transmission bandwidth. In the proposed system, the input light undergoes intensity modulation (OOK), which encodes communication data as an optical carrier for propagation along the optical fiber. As light is subjected to perturbation, the intensity/phase of the carrier is further modulated. At the detection stage, digital signal processing can separate communication data (high frequency) and vibration signals (low frequency) for individual processing. Information on the vibration amplitude, frequency and position can be extracted through signal analysis.

The proof-of-concept demonstration of the integrated system combining sensing and communication is shown in Figure 2. The system comprises two parallel but counterpropagating optical transmission paths, including identical transmitter and receiver modules. The starting point of each optical path consists of a laser source, which emits linearly polarized continuous-wave light at 10 mW, with a wavelength of 1550.12 nm and a linewidth of 100 Hz. A narrow linewidth laser was used for two main reasons. For sensing, narrow linewidth lasers can provide better phase and intensity stability, which facilitates a higher signal-to-noise ratio and thus allows for higher-precision measurements. For communication, especially for long-distance fiber-optic transmission, different frequencies of light propagate at different speeds due to chromatic dispersion, which can distort the signal transmitted. Using a narrow linewidth laser can reduce the impact of dispersion. At the same time, such lasers are less noisy and can contribute to an adequate signal-to-noise ratio.

A pre-programmed pseudo-random bit sequence generated by an arbitrary waveform generator (AWG) is converted to an analog voltage signal which, after voltage amplification, drives an electro-optic modulator (EOM) to facilitate transmission or non-transmission of light as an intensity modulator. The modulation rate of the optical signal is 150 Mb/s (OOK format). The intensity-modulated light is injected into a single-mode telecom fiber (G.652) of approximately 100 km. A 60 m length fiber is fixed around a cylindrical piezoelectric transducer (PZT) to induce external vibrations. The PZT deformation and strain transfer to the optical fiber, producing intensity and phase modulation, are controlled by a signal generator and voltage amplifier. The average responsivity of the PZT is measured at 32.9 rad/V at 1550 nm, corresponding to a voltage-strain coefficient of 91.6 nε/V. At the detection end, light enters a 90° optical hybrid and interferes with an optical local oscillator. The resulting light is converted into a voltage signal by a pair of balanced photodetectors (BPD) with a bandwidth of 400 MHz. The IQ demodulation method is chosen to extract the phase information from the signal, with phase unwrap to ensure signal continuity. Information on the intensity of the signal can also be obtained from the square root of the sum of I and Q powers (AC coupled), which is proportional to the total optical power received. However, the main contributor to intensity modulation is not from vibration-induced bend loss, but from the mismatch between the states of polarization of the interfering light, which changes the interference visibility and thus the proportionality factor. A real-time oscilloscope with a bandwidth of 3 GHz and a sampling rate of 1.25 GS/s (communication demodulation) or 12.5 MS/s (sensing demodulation) is used. Note that a down-sampling factor of 100 is employed for real-time sensing data processing, which has lower hardware requirements.

The vibration position can be calculated from the time delay (Δ*t*) between the signals received by each demodulation end: (1)$$Z_{m}=\frac{1}{2}\left(L+\frac{c∆t}{n}\right),$$
where c is the speed of light in a vacuum, *L* is the length of the sensing fiber and n is the effective index of the fiber core.

## 3. Results and Discussion

### 3.1. Sensing Operation

The sensing performance was characterized in the absence of any communication data being transmitted. To evaluate the impact on the optical intensity and phase under different strains, a series of measurements was carried out, as shown in Figure 3. A linear relationship was observed between the applied strain and the intensity change at a vibration frequency of 1 kHz, indicating a sensitivity of 1.08 μW/με. The corresponding LoD (under a bandwidth of 400 MHz) is 33.76 pε/Hz^1/2^.

It is worth noting that phase demodulation is superior to intensity demodulation in terms of vibration positioning accuracy, and is thus adopted for the majority of the experiments in this series of demonstrations.

To evaluate the frequency response, the sensitivity and corresponding limit of detection (LoD) at different vibration frequencies were obtained and compared. It can be seen in Figure 4 and Table 1 that with an increase in frequency from 10 Hz to 1 kHz, the sensitivity remained consistent, but there is a decreasing trend for LoD. This can probably be attributed to the well-known phenomenon of signal fading, which decreases the signal-to-noise ratio and appears to be stronger at higher frequencies. The range of frequencies tested were limited to the available PZTs in the experiment.

Along a 101.27 km length optical fiber, after demodulating and obtaining the phase signals at both detection ends, the cross-correlation algorithm was used to calculate the time delay between the signals and the vibration position. Figure 5a displays the demodulated phase signal after applying a single-point vibration with a PZT at 101.24 km. The resulting measurement after signal processing yields an average vibration position of 101.26 km, which is in excellent agreement with the actual position obtained from an incoherent optical time-domain reflectometer. Similarly, when a single-point vibration is applied at 50.63 km, the measured vibration position is 50.60 km.

However, when multiple vibrations occur along the sensing fiber, the cross-correlation algorithm is unable to resolve the individual positions. To address the scenario of multiple vibrations, which is likely in a real measurement environment, the phase spectrum time delay method [21] for vibration positioning was adopted. Specifically, cross-correlation is performed on the phase signals, and then Fourier transform is used to obtain the phase spectrum of the signals. The time delay is then analyzed, and thus the vibration position. Note that this method, or any other method for forward-transmission distributed sensing, cannot distinguish the vibration influence length, nor can it spatially resolve multiple identical-frequency vibration sources. In this case, 150 Hz and 100 Hz vibrations at 50.63 km and 101.24 km, respectively, were applied simultaneously. The demodulated phase signals are shown in Figure 5b. From the phase spectrum presented in Figure 5c, it is evident that the 150 Hz and 100 Hz phase signals are clearly detectable. Using the phase spectrum time delay method, the corresponding time delays for these two frequencies are 0.028 ms and 0.499 ms, respectively, as shown in Figure 5d. These time delays correspond to vibration positions of 51.64 km and 101.69 km, respectively. This demonstrates that the positioning of multiple vibrations is feasible at the basic level.

To evaluate the positioning accuracy of vibration sensing, 60 repetitive measurements were conducted under a communication–sensing integration scenario with both data transmission and vibration-induced perturbations, which are plotted in Figure 6. The standard deviation (STD) of the positioning results was calculated to be 10.10 m, indicating a relatively tight clustering of the measurements around the mean value. This suggests that the system exhibits a high degree of repeatability and predictability in its positioning capability. Another key metric for assessing positioning accuracy, the root mean square error (RMSE), was also computed in order to quantify the typical magnitude of the positioning errors. With a RMSE of 29.02 m, the findings suggest that while there is room for improvement towards achieving sub-meter precision, the existing system is capable of delivering reasonably accurate position estimates for most large-scale applications.

The spatial resolution from using the phase spectrum time delay method [21] was found to be as small as 0.08 m, when considering the spatial resolution in the frequency domain (which depends on phase resolution) rather than the time domain (which depends on the sampling rate): d*Z* = *c*/(2 × *n* × *N* × *f*) [22], where *n* is the effective index of the optical fiber (e.g., 1.467 @ 1550 nm), *N* is the number of sampling points per measurement (e.g., 1.25 M) and *f* is the signal frequency (e.g., 1 kHz). However, the spatial resolution cannot be infinitely enhanced by increasing the number of data points, as hardware limitations and the measurement rate requirement require a lower limit. It is interesting to note that if the conventional method based on the sampling rate (12.5 MS/s) was used, the spatial resolution would only be 16.4 m.

### 3.2. Communication Operation

In the transmission of communication data, upon receiving an optical signal, the corresponding voltage level needs to be sampled and the binary value decided. A decision threshold must be set, which is typically positioned between the anticipated logic levels “0” and “1”. For OOK, “0” corresponds to “low” optical power, while “1” corresponds to “high” optical power. A nominal voltage of 70 mV serves as the decision threshold. If the voltage sampled is above 70 mV, the signal is determined to be “1”; otherwise, it is determined to be “0”. The communication data recovered through this method are presented in Figure 7. When vibrations occur, the OOK-encoded message (optical carrier) experiences visible phase fluctuations, which fortunately do not pose a demodulation problem. Meanwhile, the intensity is also affected by the perturbations, resulting in changes to the extinction ratio of the optical carrier. However, as long as the fluctuation amplitude does not prevent the demodulated signal from crossing the threshold, it has a negligible impact on data recovery.

After recovering the communication signal, the quality of the signal is analyzed to help evaluate the performance of the system. The Q-parameter is commonly used to describe signal quality, which is a measure of the ratio of signal level to noise level. For binary signals, the Q-parameter can be calculated using the following formula:(2)$$Q=\frac{V_{1}-V_{0}}{\sigma_{1}+\sigma_{0}},$$
where $V_{1}$ and $V_{0}$ are the average values of the high-level and low-level signal voltages, and $\sigma_{1}$ and $\sigma_{0}$ are the standard deviations of the noise of the high-level and low-level signal voltages, respectively. A larger Q-parameter indicates better communication quality. The Q parameter on the decibel scale is converted from the linear scale via the following relationship:(3)$$Q_{dB}=20\times {log}_{10}(Q).$$

The relationship between the well-known bit error rate (BER) and the Q-parameter can be approximated by the following expression:(4)$$BER=\frac{1}{2}\left[erfc\left(\frac{Q}{\sqrt{2}}\right)\right]$$

The correspondence between $Q_{dB}$ and BER in engineering is shown in Table 2. It can be seen that, as the Q value increases, BER increases exponentially.

By plotting an eye diagram as shown in Figure 8, one can visually observe the impact of disturbances along the fiber path on the communication signal. With vibration-induced noise applied by a PZT, phase noise from the laser source and from the ambient environment and various noise contributions from the BPD and oscilloscope, the high and low voltage levels are not perfectly stable on a fixed horizontal line. The mean of the high voltage level, representing a “1”, is 127.32 mV, with a STD of 18.34 mV. Similarly, the mean of the low voltage level, representing a “0”, is 12.42 mV, with a STD of 4.33 mV. Using these data, the *Q_dB_* of the communication system observed within this timeframe is 14.08 dB, which indicates that the BER is 2.8 × 10^−7^.

### 3.3. Integration Impact Analysis

In OOK communication, only the intensity of light is modulated at the transmitter end, at a relatively high frequency. Vibration sources encode their information into the propagating light in the form of low-frequency phase, intensity and polarization changes. The communication and sensing signals at the receiver end can be separated for independent analysis due to their different frequencies, as well as different demodulation schemes (intensity/phase). Sensing involves low-frequency phase demodulation, while communication involves high-frequency intensity demodulation.

The extent of crosstalk between communication and sensing signals is of paramount importance to an integrated system. A comparative study was undertaken to investigate the impact of vibrations on the quality of data transmission, and, vice versa, the impact of data encoding on the detection of vibration signals. Figure 9 reveals the outcome from four series of experiments that systematically explored the on/off situation of either the EOM (communication) or PZT (vibration sensing). Figure 9a shows system noise in the absence of both high-frequency intensity modulation (OOK) and low-frequency intensity modulation (vibration). Figure 9b shows the detection of sinusoidal vibration signals without OOK modulation, with an LoD of 8.28 pε/Hz^1/2^ and a positioning accuracy (STD) of 9.36 m. Figure 9c shows OOK modulation without vibration perturbations, which reveal an extinction ratio of 6.57 dB, *Q_dB_* of 17.05 dB and BER of 7.4 × 10^−13^. Lastly, Figure 9d contains both OOK modulation and vibration signals, which exhibit an extinction ratio of 7.62 dB (trough of vibration envelope) or 8.52 dB (peak of vibration envelope), a *Q_dB_* of 14.08 dB and a BER of 2.8 × 10^−7^. The sensing aspect possesses an LoD of 7.56 pε/Hz^1/2^ and a positioning accuracy of 10.10 m.

It is clear that communication has a negligible impact on the sensing of vibrations, as shown by comparing the LoD (OFF 8.28 pε/Hz^1/2^ vs. ON 7.56 pε/Hz^1/2^) and the STD (OFF 9.36 m vs. ON 10.10 m) in Figure 9b,d. As for the impact of vibrations on communication, the impact is relatively minor (as evidenced by the smooth operation of commercial communication systems), as can be seen by comparing the extinction ratio (OFF 6.57 dB vs. ON 7.62–8.52 dB), the *Q_dB_* (OFF 17.05 dB vs. ON 14.08 dB) and the BER (OFF 7.4 × 10^−13^ vs. ON 2.8 × 10^−7^) in Figure 9c,d. The considerable improvement in the extinction ratio when vibrations are present can be attributed to a change in the state of polarization and the effect of polarization-dependent loss (PDL). However, this does not indicate improved communication quality under vibration conditions. Rather, the intensity fluctuations caused by vibrations can be regarded as an increase in overall noise, which directly leads to a decrease in *Q_dB_*.

Although the bit rate (150 Mb/s) demonstrated in this study is low compared to that found in communication-oriented fiber-optic systems, the goal is to explore the feasibility of deep integration between communication and sensing. A higher bit rate would require dedicated hardware, although the concept and signal processing methods would remain the same; thus, this approach to deep integration should remain feasible for high-speed large-capacity communication systems.

## 4. Conclusions

We reported on the deep integration between a fiber-optic communication system employing OOK-encoded data transmission and a type of forward-transmission-based distributed fiber-optic vibration sensor using intensity demodulation. Both functionalities are realized using the OOK electro-optical system, comprising a laser source, AWG, EOM, IQ coherent receiver and computer. An increase in the BER from 7.4 × 10^−13^ to 2.8 × 10^−7^ was observed for a bit rate of 150 Mb/s in the presence of vibrations. Correspondingly, the *Q_dB_* decreased from 17.1 to 14.1. It must be stressed that the impact on communication is not caused by the addition of non-intrusive sensing, but by the vibration-induced noise already present in data transmission. Likewise, no significant impact was seen in the sensing performance metrics when communication data were transmitted through pulse code modulation. The single-span sensing distance is 101.3 km (no optical amplification), the spatial resolution is 0.08 m, positioning accuracy can be as low as 10.1 m, the sensitivity is 399.0 rad/με, the LoD is 8.3 pε/Hz^1/2^ at 1 kHz, and the vibration frequency range tested is 10–1000 Hz. No data averaging was performed during signal processing; this reduced the computation power requirement. This proof-of-concept demonstration shows that deep integration (share laser, optical fiber, optical signal and demodulator) between fiber-optic communication and distributed fiber-optic sensing is possible while adequately preserving the quality of dual functionalities.

Potential users of this technology could include telecommunications companies, seismology agencies and governments, all of whom could benefit from the potential cost savings. Since no hardware modifications are required, the cost of deploying this technology would mainly relate to software services (to cover development and maintenance costs). Such costs should be much lower than purchasing physical equipment (conventional DAS costs around USD 100–500 k) and deploying dedicated new optical cables (typically USD 30–50 k per km). It is anticipated that the operating cost of using the proposed technology to provide coverage for an existing optical cable link will not exceed USD 100 k per year.

## Figures and Tables

**Figure 1 sensors-24-05758-f001:**
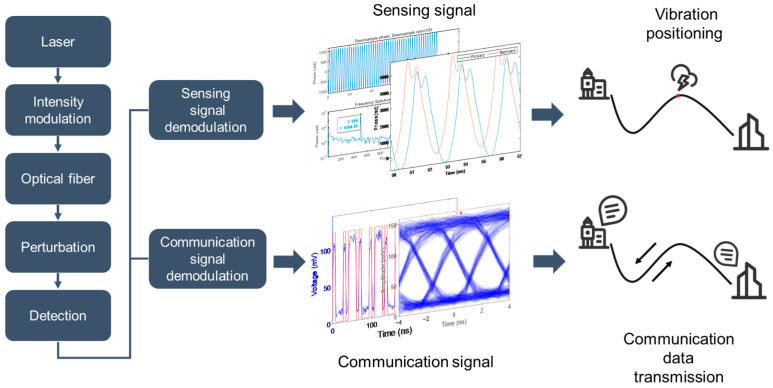
Concept of deep integration of fiber-optic communication and sensing systems.

**Figure 2 sensors-24-05758-f002:**
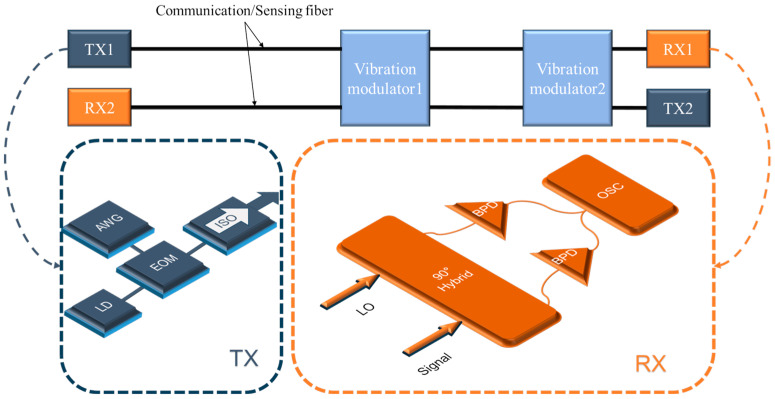
Experiment setup of the integrated system. TX: transmission module, RX: receiver module, TLD: laser diode, EOM: electro-optic modulator, AWG: arbitrary waveform generator, ISO: isolator, BPD: balanced photodetector, OSC: oscilloscope.

**Figure 3 sensors-24-05758-f003:**
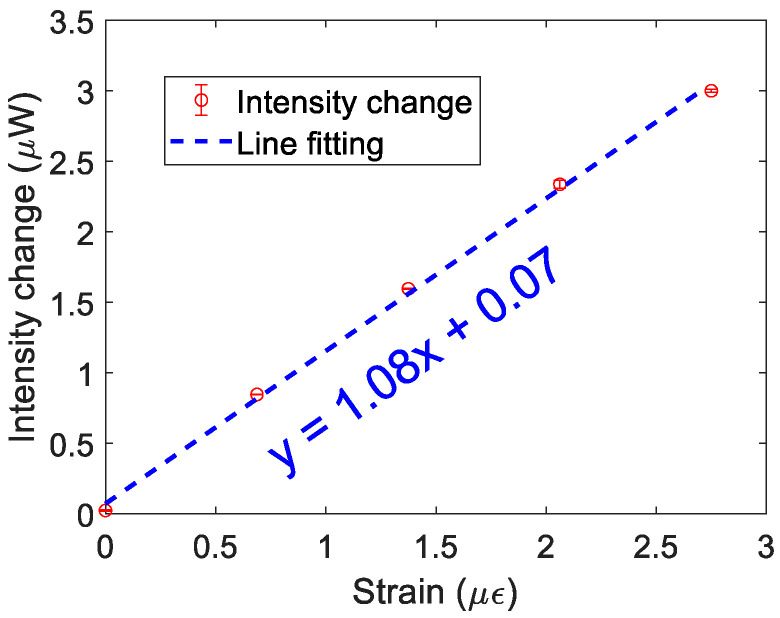
Intensity demodulation: relationship between intensity change and applied strain.

**Figure 4 sensors-24-05758-f004:**
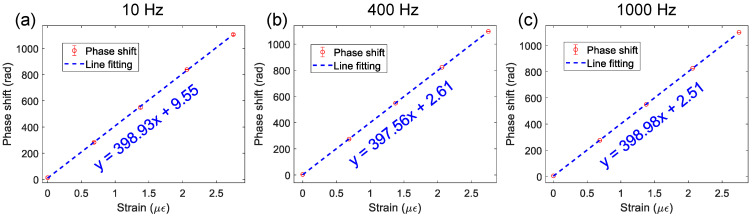
Phase demodulation: relationship between phase shift and fiber strain. (**a**) Sensitivity at 10 Hz; (**b**) sensitivity at 400 Hz; (**c**) sensitivity at 1 kHz.

**Figure 5 sensors-24-05758-f005:**
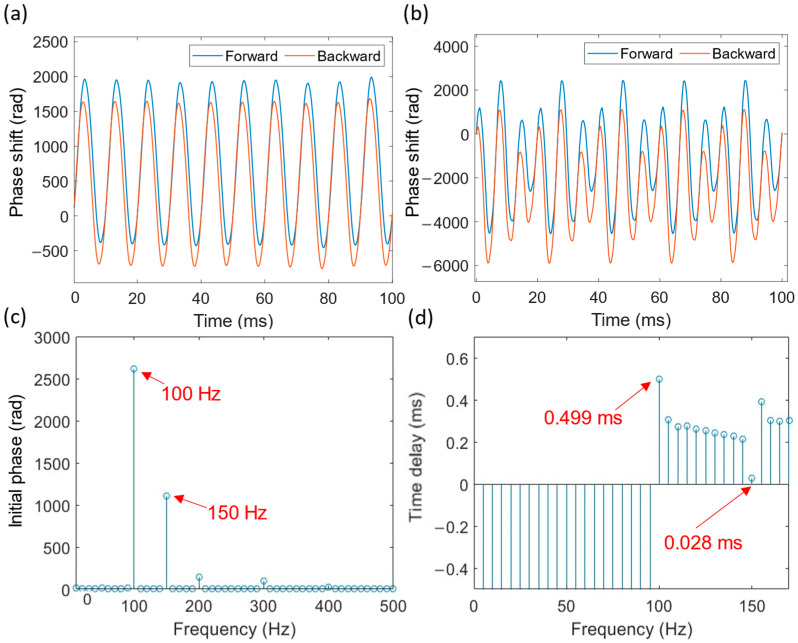
Phase demodulation: vibration positioning. (**a**) Phase signal of single-point vibration; (**b**) phase signal of two-point vibration; (**c**) phase spectrum of Fourier transform; (**d**) phase spectrum time delay.

**Figure 6 sensors-24-05758-f006:**
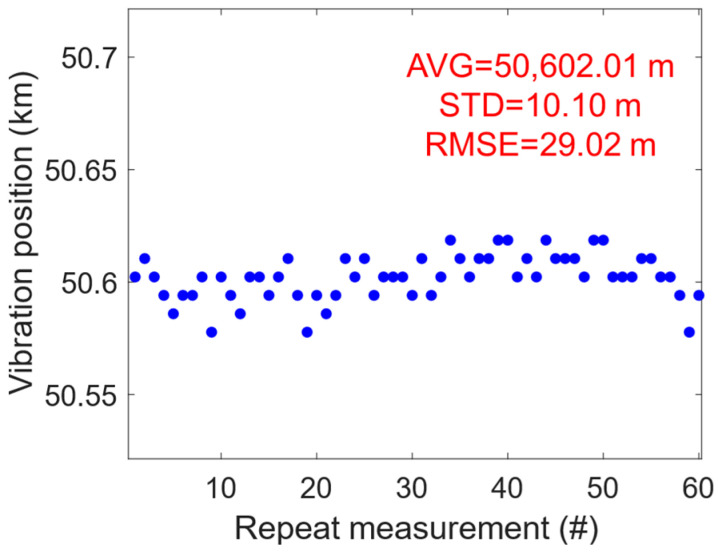
Phase demodulation: positioning accuracy through repetitive measurements.

**Figure 7 sensors-24-05758-f007:**
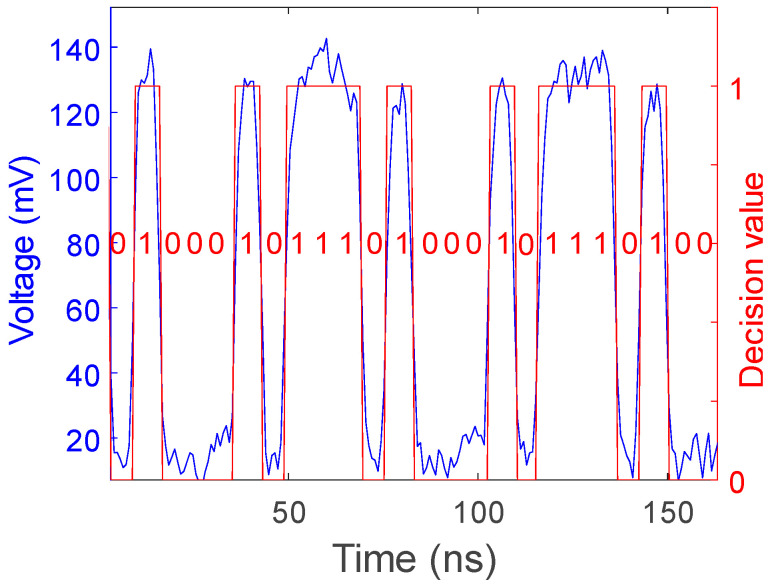
Intensity demodulation: bit-sequence recovery.

**Figure 8 sensors-24-05758-f008:**
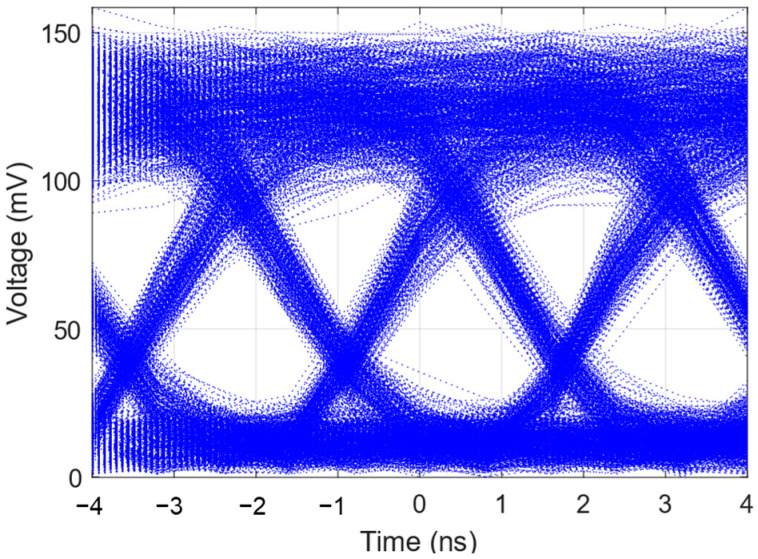
Intensity demodulation: eye diagram of on–off keying (OOK) data transmission.

**Figure 9 sensors-24-05758-f009:**
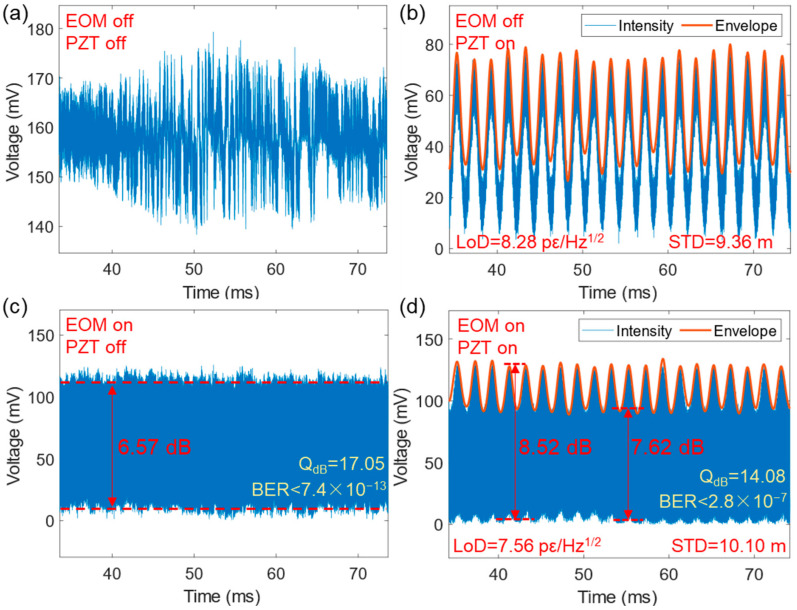
Communication (intensity-)sensing (phase) crosstalk analysis. (**a**) No data transmission and vibration; (**b**) vibration only; (**c**) data transmission only; (**d**) data transmission with vibration. Applied vibration amplitude is 2.75 με.

**Table 1 sensors-24-05758-t001:** Sensitivity and limit of detection (LoD) at different vibration frequencies.

Frequency (Hz)	Sensitivity (rad/με)	LoD (pε/Hz^1/2^)
10	398.93	21.38
400	397.56	15.49
1000	398.98	8.28

**Table 2 sensors-24-05758-t002:** Correspondence between *Q_dB_* and bit error rate (BER).

** *Q_dB_* **	13	14	15	16	17	17.1	17.2
**BER**	4.2 × 10^−6^	2.8 × 10^−7^	9.6 × 10^−9^	1.4 × 10^−10^	7.4 × 10^−13^	4.1 × 10^−13^	2.2 × 10^−13^

## Data Availability

Data underlying the results presented in this paper can be made available upon request.

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
