# Peer review of "Deep Integration of Fiber-Optic Communication and Sensing Systems Using Forward-Transmission Distributed Vibration Sensing and on–off Keying"

_sensors, 2024, doi:10.3390/s24175758_

Round 1

Reviewer 1 Report

Comments and Suggestions for Authors

    The authors propose a proof-of-concept demonstration between forward-transmission distributed fiber-optic vibration sensing and on-off keying (OOK) based communication system, and analyze the performance and test results respectively. The combination of sensing and communication is a developing trend, and the author's work is very meaningful. As I see, the technical advance and measurement results presented in this version should be further modified before publishing.

1. The linewidth of the light source is 100Hz, will this affect the data transmission of the communication? Or, what is the impact of the line width of the light source on communication and sensing, and what is the basis for the line width selection.

2. OOK coding is mainly based on communication, but in the analysis of sensing data, what is the impact of this coding on sensing? How to decode?

3. Phase demodulation is used in this manuscript. Why demonstrates Figure 3? Or, Fig. 3 can use two diagrams to illustrate the advantages of phase demodulation.

4. As the authors described in the end of Page 5, the position result is in excellent agreement with the actual position obtained from an optical time-domain reflectometer. Testing with traditional OTDR? Or phase OTDR? It should be described clearly.

5.The standard of English expression and writing of scientific papers should be improved. For example: Ref.[12] and Ref.[13], ‘’Ieee’’ should be written as ‘’IEEE’’. Moreover, the format of author names is also inconsistent.

Comments on the Quality of English Language

The English expression should be improved to clearly and easy to understand.

Reviewer 2 Report

Comments and Suggestions for Authors

This is a fascinating paper, which shows how to use already-installed fiber optic cables to sense a distant earthquake or vibrations from similar sources, such as nuclear explosions. Installing the recommended equipment could constitute an early warning system for such events. The authors point out that no degradation of information transmission in the cables will occur. And the equipment permits localizing the event within 29m. This is a great case for adding functionality without cost to the primary system. However, one must ask how many people will pay for this capability. To make it more attractive the authors should tell us what an installation will cost. I don't think it will be very much. It operates simply by timing the time delay between signals detected at each end of the cable.
